# Segmenting Olive Oil Consumers Based on Consumption and Preferences toward Extrinsic, Intrinsic and Sensorial Attributes of Olive Oil

Anita Silvana Ilak Peršurić 

Department Economics and Rural Development, The Institute of Agriculture and Tourism, 52440 Poreč, Croatia; anita@iptpo.hr; Tel.: +385-052-408-329

**Abstract:** The aim of this paper was to identify and describe segments of a study population that consumes olive oil. Therefore, a survey was conducted in 2019 on a sample of 705 German and 175 UK consumers. In both samples, three consumer segments were identified. These three segments differed significantly with regard to purchase and consumption patterns, as well as attitudes toward the extrinsic, intrinsic, sensorial, and health attributes of olive oil. Their main preferences related to health properties of olive oil, followed by hedonic attributes; therefore, these aspects should be marketed in German markets. For UK consumers, validation of extrinsic attributes, such as region, micro-location, and protected designation of geographical origin (PDO), should be used in marketing campaigns in light of economic sustainability on local family farms and small and medium-sized enterprises (SMEs). Since UK consumers rely heavily on vendors' recommendations, more effort should be made in the UK market to establish habits of consumption and an olive oil culture that would be sustainable in long-term time frames.

**Keywords:** consumers; segments; olive oil; attributes; Germany; UK

---

## 1. Introduction

The agricultural market is under continual pressure from global economic and environmental demands to address to changing demands, habits, and purchase and consumption patterns of consumers around the world. Correspondingly, market surveys that focus on consumers and their behavior toward certain agricultural products are widely used by producers and marketers who seek to satisfy consumers' needs in an appropriate way. The olive oil market is no exception; it complies with the same rule. The recent surveys about olive oil prevail in leading producing countries, such as Italy [1,2], Greece [3], and Spain [4], as well as a few about other markets like Argentina [5], Chile [6], Brazil [7], Germany [8], or the UK [9].

Several authors support the hypothesis that olive oil consumption can be increased by analyzing consumers' preferences, especially in countries where olive oil is a non-traditional product, like Argentina [5], Chile [6], and the UK [9], due to its external attributes and hedonic and health dimensions, which prove to be highly important in its rejection or acceptance.

The idea that, among the general population of consumers, there are those who exhibit different preferences toward olive oil and express different expectations of its attributes led researchers to create surveys and to examine in which way socio-demographic features of consumers affect their behavior (Table 1). In Italy, the segmentation of [1] showed that the expert group gave more importance to intrinsic attributes compared to utilitarian and naïve consumers. Additionally, in Italy, two clusters were identified according to their preferences for color, flavor, and taste of olive oil [10], while consumers' knowledge about the meaning of organic olive oil was referred by [2]. In Spain, upper social groups

and higher income groups were more aware of protected designation of geographical origin (PDO) labels [11]. Even religious connotations of olive oils (produced in pilgrimage countries) have proven to be important in consumers' segmentation [12]. In several studies about olive oil attributes, the concept of country of origin was highlighted as an important feature for olive oil and other products, such as French wines and cheeses, German beer, Swiss and Belgian chocolate, and Italian wine, cheese, and pasta [13]. Moreover, recent studies on food products also showed the importance of country of origin, suggesting that a certain product image reflects the image of the region or country of production [13–15].

**Table 1.** Socio-demographic features of olive oil consumers in surveys.

| Authors | Gender Female(F) Male (M) | Age (Years) | Education | Income/Occupation | Country |
|---|---|---|---|---|---|
| [3] | 64.7% F 35.3% M | 31–50 (45.7%) | 58.6% University | 1001–2000 EUR in majority | Greece |
| [4] | 52.7% F 47.0% M | 20–39 (34.0%) 40–54 (35.0%) ≥55 (31.0%) | University (27.4%) | ≤1000 EUR (19.2%) 1000–2000 EUR (47.4%) 2001–3000 EUR (21.9%) ≥ 3000 EUR (11.5%) | Spain |
| [6] | 49.0% F 51.0% M | 18–34 (38.0%) 35–49 (34.0%) >50 (21.0%) | Secondary (46.0%) Postsecondary (42.0%) Post graduate (12.0%) | ≤1500 USD (73.0%) 1500–2000 USD (18.0%) >2000 USD (9.0%) | Chile |
| [7] | 46.0% F 54.0% M | ≤18 (2.0%) 19–30 (4.0%) 31–50 (41.0%) >50 (16.0%) | Secondary (31.0%) University (54.0%) Post doc (27.0%) | - | Brazil |
| [12] | 100% F | 20ties (16.3%) 30ties (17.5%) 40ties (25.7%) 50ties (20.3%) 60ties (20.2%) | Junior high school (1.3%) Senior high school (25.3%) Junior college (20%) University (34.4%) Master/PhD (1.9%) Professional/technical school (16.2%) Other (1.1%) | - | Japan |
| [16] | 61.5% F 38.5% M | 20–29 (33.3%) 30–39 (16.0%) 40–49 (16.0%) 50–59(14.4%) ≥60 (20.3%) | Primary (9.9%) Middle (17.6%) Secondary (52.2%) University (20.2%) | Employed (41.3%) Unemployed (4.5%) Retired (19.2%) Housewife (12.2%) Student (22.2%) | Italy |
| [17] | 51.0% F 49.0% M | Average 39.1 years | Lower than high school (17.9%) High school (45.4%) College/university (36.7%) | ≤1000 EUR (4.6%) 1000–2000 EUR (45.4%) 2001–3000 EUR (28.6%) 3001–4000 EUR (6.6%) ≥4001 EUR (10.7%) | Italy |

For Spanish oils, attributes of origin, region of production, and quality directly affect their market potential [11], while for Italian consumers olive oil production region is an important driver of choice [18]. In Italy, consumers showed very high knowledge and demand for extrinsic attributes, such as place of production, designation of origin, organic certification, and type of processing for extra virgin olive oils [16], which are a link between production sustainability and perception of high quality.

Cultural differences in olive oil consumption habits were identified as crucial for the consumers in the Mediterranean cultural surrounding (even different cultures in the same country evaluate and purchase food in different ways), e.g., French consumers value the country of origin, while Tunisian consumers valued the region of origin and olive variety [19].

The attribute of color was surveyed in Tunis and France by [19] on two different types of oils, a deep green one and gold yellow one; Tunisian consumers were more aware of quality and valued more of a green color. In Spain, the yellow color of the Arbequina variety was more appealing to consumers in the

region of Terragona, while the deep green color was more appreciated in Andalucia [20]. In Uruguay, green and deep green colors were perceived as expensive, richly flavored, and tasty oils [10]; the pale yellow color was perceived as being of poor quality, cheap, and having a milder taste, and was therefore least acceptable.

Knowledge about olive oil also proves to be crucial in purchasing, since consumers in regions that produce olive oil have higher levels of knowledge and acceptance of olive oil in their diet, as well as more positive attitudes toward extra virgin and organic olive oil [5]. On the other hand, a lack of knowledge, e.g., about organic olive oil, leads to poor differentiation on the market and no additional market value [2].

Packaging of olive oil was evaluated as a utility; therefore, plastic bottles were perceived as more practical for everyday use by UK consumers. Furthermore, olive oil was perceived as cooking oil by UK consumers [9].

Health properties were most important to olive oil consumers in Greece [21], and were most important to the segment of high-income, highly educated, and married persons above 40 years of age. According to [21], the health aspects of olive oil were more important to women, while the higher involvement group rated the evaluated attributes higher. These findings also mention the protection of the environment, economic advantages, maintenance of the social structure, and protection of consumers' and producers' health.

Sustainability in olive oil production can be achieved for small farmers due to traditional processing techniques (milling using stone mills, hand picking), which are highly valued by consumers [22] and might improve economic sustainability for small farmers. For farmers and local producers, the local consumption and purchasing are increasing the economic sustainability on the micro-location level, while in the wider discourse, they enable the sustainable development of rural areas [23–25]. Sustainability also relies on consumers' behavior toward local production and purchase of local food; these are considered to be more sustainable ways of trading (similarly to wine in [26]).

The purpose of this paper was to find out how German and UK olive oil consumers differ according to their consumption habits and preferences of olive oil attributes and to search for subgroups or segments within the group. The findings of this survey might improve the attributes of olive oil according to consumers' preferences. Since consumers are more aware of and more demanding toward extrinsic and intrinsic attributes of products, this survey may also help sales and marketing specialists. In order to satisfy consumers' needs, olive oil producers may use these research results as well as the example of German and UK olive oil consumers' demands. Finally, according to our findings, possible economic sustainability effects might be reached through the enhanced volumes of olive oil sales to tourists and visiting persons in Croatia, either on local farms or in lodging facilities, becoming a part of the tourist system.

## 2. Materials and Methods

### 2.1. Data Collection

Data were gathered in 2019 during the period from June to September on a sample of people from Germany and the UK who were staying in Istria, Croatia during this period. Specifically, the survey was conducted in the following cities: Novigrad, Poreč, Rovinj, and Umag. We made agreements with the hotel companies beforehand, apprising them of our survey's intent and posting an associated announcement at these locations. According to the predetermined schedule, we visited one hotel each day in the aforementioned cities. We used the method of random sampling among German and UK tourists who were present in the hotels on the scheduled days. Participants were of different socio-demographic and economic backgrounds, had different educations, ages, and living standards, and ranged from unemployed, employed, to retired. Participants in the survey were approached in their hotels by trained researchers (including the author of this paper) and asked if they would like to participate in the survey. The purpose of the survey was explained, as was the fact that it would be

anonymous. While the participants filled out the questionnaires, the researchers were available for the entire time to address eventual questions or remarks or to provide additional information. In total, 880 valid questionnaires were collected and statistically processed.

## 2.2. Sampling Tool—The Questionnaire

The sampling tool for data collection was a questionnaire that contained both open and closed types of questions. This questionnaire was previously tested on a convenient sample of 35 Croatian consumers of olive oil in order to test its clarity and understanding and to test the statistical possibilities of the gathered data. After textual and technical adjustments, it was translated into the German and English languages by a certified translator.

The questions about olive oil quality attributes were selected after consultation with the head of the Olive Oil Sensorial Tasting Panel (Panel) at the Institute of Agriculture and Tourism in Poreč, Croatia (of which the author is a member) and in consultation with the German virtual panel (which came to visit the institute's Panel for cross-country tasting and experience sharing). On the part of the Croatian Panel, all standard descriptive parameters for olive oil were used (IOC standard [27]), while on the part of the German Panel, the knowledge about the preferences of German olive oil consumers and olive oil market was used. Based on a verbal communication with several members of the German Panel who worked for established German companies (importers and sales), laboratories, and state authorities, we found out that German olive oil consumers prefer a hint of sweetness in taste and a low intensity of green notes, green leaves, grass, and olives in smell and taste.

The professional grading for the sensory properties of olive oil according to the IOC and Panel standards use a scale of 1 to 10, while for the purpose of the questionnaire and our survey, a more appropriate Likert scale was used (the scale was graded from 1 to 5, with 1 as not important and 5 as an extremely important attribute). Extrinsic (origin, package, price) and intrinsic (expert opinions, PDO, sensory) attributes of olive oil were also rated using a Likert scale from 1 to 5 based on [26].

The knowledge about olive oil was described by perceived knowledge or self-knowledge based on so-called psychographic characteristics [6]. In order to measure the judgments of self-estimated knowledge, we also used a Likert scale from 1 to 5.

Olive oil consumption was measured in terms of monthly and annual expenditure in liters and euros. The frequency of consumption was measured from "I do not consume at all" to "Multiple times a day". The habits of olive oil consumption were described through consumption at home, in restaurants, while traveling, and other ways (open answer).

The sociodemographic features of participants and their habits of consumption were described in Table 2.

**Table 2.** Socio-demographic features of participants in the survey.

| German N = 705 | | | | |
|---|---|---|---|---|
| **Gender** | **Age (Years)** | **Education** | **Occupation** | **Monthly Personal Income** |
| F (63.3%) M (36.7%) | 20–30 (13.9%) 31–40 (21.9%) 41–50 (26.2%) 51–60 (24.6%) ≥ 61 (13.4%) | Elementary (8.7%) Secondary (46.4%) University (44.9%) | Entrepreneur (5.4%) Self-employed (5.7%) Manager (9.3%) Employee (53.5%) Retired (8.4) Pupil/student (3.4%) Unemployed (0.9%) Other (e.g., rentiers, housewives) (13.4%) | ≤1000 EUR (14.9%) 1001–2000 EUR (34.0%) 2001–3000 EUR (27.9%) 3001–4000 EUR (10.6%) ≥4001 EUR (12.6%) |

**Table 2.** *Cont.*

| UK N = 175 | | | | |
|---|---|---|---|---|
| **Gender Female(F) Male (M)** | **Age (Years)** | **Education** | **Occupation** | **Monthly Personal Income** |
| F (42.3%) M (52.7%) | 20–30 (16.7%) 31–40 (25.0%) 41–50 (30.6%) 51–60 (16.0%) ≥61 (11.8%) | Elementary (0.7%) Secondary (36.7%) University (62.6%) | Entrepreneur (4.1%) Self-employed (16.3%) Manager (21.1%) Employee (47.6%) Retired (0.0) Pupil/student (6.8%) Unemployed (2.7%) Other (e.g., rentiers, housewives) (1.4%) | ≤1000 EUR (11.9%) 1001–2000 EUR (16.9%) 2001–3000 EUR (29.9%) 3001–4000 EUR (22.4%) ≥4001 EUR (19.4%) |

*2.3. Sample Description*

*2.4. Methods of Data Processing*

Confirmatory factor analysis was applied to test the factor solutions and statistical robustness of variables relying on each factor of the matrix [28]. The suitability of the data was assessed using the Kaiser–Meyer–Olkin measure of sampling adequacy and the Bartlett test of sphericity. The data were processed by factor analysis with maximum likelihood estimation and varimax rotation with Kaiser normalization. Cross-loadings were limited to one factor by limiting item loadings above 0.4 and associated with only one factor in the factor matrix. This level of factor loadings indicated stable factors in the matrix [28,29].

Cluster analysis was chosen for consumer segmentation to uncover similarities within the groups of olive oil consumers. We supposed that we can establish the hypothesis that groups will have some properties in common regarding consumption, purchase, and extrinsic, intrinsic, and sensorial attributes of olive oil.

We used a non-hierarchical technique (K-mean clustering) because of the a priori hypothesis that the group of olive oil consumers will contain different sub-groups or clusters (similarly to [30]). The mean values of attributes method was used [30]. The clustering criteria were olive oil consumption and oil attribute validation. The obtained consumer segments were interpreted using the original variables of consumption behavior, habits, and attitudes toward extrinsic, intrinsic, health, and sensorial attributes of olive oils and perceived knowledge or self-knowledge about olive oil.

**3. Research Results**

*3.1. Factor Analysis*

The factor analysis resulted in three factors that created the factor matrix of consumption behavior toward olive oil (Table 3): Consumption and purchase habits (Factor 1), the health properties of olive oil (Factor 2), and the general and perceived self-estimated knowledge about olive oil (Factor 3).

Consumption and purchase habits were the first factor with highest factor loadings in the factor matrix, with consumption frequency having with highest score (0.857), closely followed by annual consumption (0.805) and monthly expenditure expressed in euros (0.460).

The health aspects of olive oil were very highly valued by German consumers, as they perceived that olive oil improves the blood vessels' health (0.960) and heart health (0.857), and that it lowers the cholesterol in the blood (0.801). They considered it to be a natural product (0.761), believing it to be beneficial to one's health (0.598), and related it to benefits for their overall health (0.473). German olive oil consumers perceived their knowledge to be higher than the knowledge of the average consumer (0.839), and they considered themselves to know a lot about olive oil (0.852).

**Table 3.** Factor analysis table—consumption, healthiness, and knowledge of olive oil—Germans.

| German Olive Oil Consumers | Factor Loading | % of Variance |
|---|---|---|
| **Factor 1—Consumption and Purchase Habits** | | 33.8% |
| Consumption frequency | 0.857 | - |
| Consumption in L/year | 0.805 | - |
| Consumption in EUR/month | 0.460 | - |
| **Factor 2—Health Properties** | | 23.3% |
| Improves blood vessel health | 0.960 | - |
| Olive oil improves the taste and smell of food | 0.860 | - |
| Improves heart health | 0.857 | - |
| Lowers cholesterol in the blood | 0.801 | - |
| Because olive oil is a natural product | 0.761 | - |
| It is beneficial to health | 0.598 | - |
| Olive oil has a positive overall effect on health | 0.473 | - |
| **Factor 3—Knowledge about Olive Oil** | | 13.3% |
| I have a lot of knowledge about olive oil | 0.852 | - |
| I know more about olive oil than the average consumer | 0.839 | - |
| I am eating a Mediterranean diet | 0.542 | - |
| Olive oil is an important part of my diet | 0.448 | - |

Kaiser–Meyer–Olkin Measure of Sampling Adequacy: 0.843. Eigenvalue F1 = 6.0; F2 = 4.4; F3 = 1.7; Sig. 0.000.

The factor analysis of UK consumers resulted in two factors that created the factor matrix (Table 4); the first factor was very dense and comprised ten variables. UK consumers appreciated the health aspects of olive oil very highly. All health properties of olive oil saturated the first factor, namely: Olive oil has a positive overall effect on health (0.845), it improves blood vessel health (0.826), it improves heart health (0.790), because olive oil is a natural product (0.741), and it lowers cholesterol in the blood (0.722).

**Table 4.** Factor analysis table—healthiness, consumption, and knowledge of olive oil—UK

| UK Olive Oil Consumers | Factor Loading | % of Variance |
|---|---|---|
| **Factor 1—Importance of health properties and consumption habits** | | 51.9% |
| Olive oil has a positive overall effect on health | 0.845 | - |
| Improves blood vessel health | 0.826 | - |
| Improves heart health | 0.790 | - |
| Because olive oil is a natural product | 0.741 | - |
| Lowers cholesterol in the blood | 0.722 | - |
| I have a habit of olive oil consumption | 0.617 | - |
| Olive oil is part of my healthy diet | 0.566 | - |
| I am eating a Mediterranean diet | 0.524 | - |
| Improves the taste of food | 0.502 | - |
| Consumption in L/year | 0.385 | - |
| **Factor 2—Knowledge about olive oil** | | 15.9% |
| I know more about olive oil than the average consumer | 0.643 | - |
| Olive oil is an important part of my diet | 0.424 | - |
| I have a lot of knowledge about olive oil | 0.417 | - |

Kaiser–Meyer–Olkin Measure of Sampling Adequacy: 0.842. Eigenvalue F1 = 6.7; F2 = 2.9; Sig. 0.000.

Knowledge about olive oil was comprised in factor two, explaining that the surveyed UK consumers perceived their knowledge to be better than that of the general population (0.643) and perceived that they knew a lot about olive oil (0.417).

Comparing the UK and German sample, we can mention that UK consumers had a slightly lower grading of health attributes, while the volume of consumed oil was substantially lower compared to that of German consumers.

In order to reveal which extrinsic, intrinsic, and sensorial attributes of olive oil were important to consumers, we conducted a factor analysis of these features.

For the German participants, the results showed that olive oil as a product has four categories of important attributes: Sensorial, extrinsic, hedonistic, and intrinsic attributes, as described in Table 5.

**Table 5.** Factor analysis table—olive oil attributes—Germans.

| German Olive Oil Consumers | Factor Loading | % of Variance |
|---|---|---|
| **Factor 1—Sensorial attributes of olive oil** | | 35.3% |
| The smell and taste of the olive fruit | 0.836 | - |
| The smell and the taste of green leaves and grass | 0.767 | - |
| Scent and flavor of aromatic herbs (e.g., rosemary, oregano) | 0.662 | - |
| The smell and the taste of green olives | 0.531 | - |
| The smell and the taste of the ripe fruit of the olive | 0.442 | - |
| **Factor 2—Extrinsic attributes— Protected designations of geographical origin PDO, labels, certificates** | | 21.7% |
| Protected designations of geographical origin/original designation (PDO) | 0.789 | - |
| Micro-localities (e.g., Poreč, Rovinj, islands, e.g., Krk, Brač) | 0.708 | - |
| Country of origin | 0.671 | - |
| Region (e.g., Istria, Dalmacija) | 0.518 | - |
| Manufacturer/brand name | 0.495 | - |
| Awards (medals) | 0.357 | - |
| **Factor 3—Hedonistic attributes of olive oil** | | 17.1% |
| Pleasant smell | 0.814 | - |
| Pleasant taste | 0.783 | - |
| Certified chemical quality (analysis, free acidity) | 0.467 | - |
| Packaging type (Glass) | 0.453 | - |
| **Factor 4—Intrinsic attributes for olive oil** | | 10.8% |
| Recommendations of vendor (producer, shop assistant) | 0.913 | - |
| Color (green) | 0.671 | - |
| Bitterness | 0.608 | - |
| Color (yellow) | 0.529 | - |
| Recommendations of friends | 0.526 | - |
| Piquancy | 0.517 | - |

Kaiser–Meyer–Olkin Measure of Sampling Adequacy: 0.883. Eigenvalue F1 = 8.8; F2 = 3.9; F3 = 2.7; F4 = 1.9; Sig. 0.000.

The sensorial attributes of olive oil were most important for German consumers, with high factor loadings in the first factor, wherein the smell and taste of olive fruit ranked first (0.836), smell and taste of green leaves and grass second (0.767), scent and flavor of aromatic herbs third (0.662) and the smell and the taste of green (0.531) and ripe olives (0.442) last.

The factor of intrinsic attributes was highly saturated by the visible labels of quality, showing that German consumers place a high value on attributes such as protected designations of geographical origin (PDO)/original designation (0.789). For the producers of olive oil in Croatia, the valuable validation of micro-localities of production (0.708), country of origin (0.671), and region of production (0.518) explains that German olive oil consumers recognize these features of local production and evaluate Croatian olive oils highly. Awards gained at competitions were also important in assuring the consumer of the product's high quality (0.357).

Since olive oil as a product contains hedonic features that rise above the utilitarian ones, the German consumers validated pleasant smell, taste, and certified chemical analysis that proves the content to be of free acids and other valuable nutritional components. The preferred type of package was glass,

which also has a hedonic aspect connected to high quality and price (while other materials like plastic and tin are connected to lower quality and price).

The last factor contained intrinsic attributes of olive oil, which also highly saturated this factor. It points out the situation when the consumers have no previous experience with a certain olive oil or hesitate with their decision; therefore, recommendations of vendors (producer, shop assistant) are welcome (0.913). They can advise a consumer who is not familiar with a certain oil or explain the oil's attributes if it was not previously consumed. The recommendations of friends were (0.526) also valuable, since their previous experiences and suggestions might help a hesitant or picky consumer to make a choice. Important intrinsic attributes were color (green), bitterness, and color (yellow), while piquancy scored the lowest importance.

Olive oil as a product had two categories of important attributes for UK consumers: Sensorial, extrinsic, hedonistic, and intrinsic attributes (Factor 1) and hedonistic and package attributes (Factor 2), as described in Table 6.

For UK consumers, the recommendations of vendors (producer, shop assistant) were most important (0.913). We may speculate that, since the self-perceived knowledge of UK consumers was estimated to be lower than that of Germans, they rely more on expert opinions in the process of olive oil purchasing. Furthermore, geographical features were very important for UK consumers, such as region of production (0.890), micro-localities (0.786), and protected designations of geographical origin/original designation (0.667). Intrinsic attributes were also comprised in the first factor; the UK consumers highly valued the smell and the taste of the ripe fruit of the olive (0.671), the smell and the taste of green olives (0.648), the smell and the taste of green leaves and grass (0.610), and the smell and taste of the olive fruit (0.603); green color (0.671) was evaluated more highly than yellow color (0.529).

**Table 6.** Factor analysis table—olive oil attributes—UK

| English Olive Oil Consumers | Factor Loading | % of Variance |
|---|---|---|
| **Factor 1—sensorial, extrinsic, and intrinsic attributesof olive oil** | | 44.5% |
| Recommendations of vendors (producer, shop assistant) | 0.913 | - |
| Region (e.g., Istria, Dalmacija) | 0.890 | - |
| Micro-localities (e.g., Poreč, Rovinj, islands, e.g., Krk, Brač) | 0.786 | - |
| The smell and the taste of the ripe fruit of the olive | 0.671 | - |
| Color (green) | 0.671 | - |
| Protected designations of geographical origin/original designation | 0.667 | - |
| Manufacturer/brand name | 0.652 | - |
| The smell and the taste of green olives | 0.648 | - |
| The smell and the taste of green leaves and grass | 0.610 | - |
| The smell and taste of the olive fruit | 0.603 | - |
| Scent and flavor of aromatic herbs (e.g., rosemary, oregano) | 0.557 | - |
| Country of origin | 0.585 | - |
| Color (yellow) | 0.529 | - |
| Recommendations of friends | 0.526 | - |
| Certified chemical quality (analysis, free acidity) | 0.473 | - |
| **Factor 2—hedonistic and package attributes of olive oil** | | 19.1% |
| Bitterness | 0.620 | - |
| Piquancy | 0.620 | - |
| Pleasant smell | 0.492 | - |
| Pleasant taste | 0.491 | - |
| Packaging size | 0.339 | - |
| Packaging Type (Glass) | 0.309 | - |

Kaiser–Meyer–Olkin Measure of Sampling Adequacy: 0.805. Eigenvalue F1 = 8.9; F2 = 3.3; Sig. 0.000.

### 3.2. Cluster Analysis

Based on our hypothesis that certain similarities within the groups of olive oil consumers will occur, we supposed that groups will have some properties in common regarding consumption, purchase, and extrinsic, intrinsic, and sensorial attributes of olive oil.

Through the procedure of cluster analysis, we identified three clusters for German olive oil consumers which were named: (1) Frequent olive oil consumers (who highly value the hedonic and health effects of olive oil), (2) moderate olive consumers (who value the utilitarian healthy and extrinsic olive oil attributes), and (3) occasional olive oil consumers (who have low interest in consumption but were health conscious). Their sociodemographic features were described through age, employment, income, while consumption patterns were described by place and volume of consumption and expenditures (Table 7). Indicators of clusters were noted in Table 8.

The Cluster 1—the frequent sensorial and hedonistic segment—had the highest proportion of all German participants in the survey (about 60%). Compared to the other two clusters, they consume olive oil most often (every day or multiple times a day), they consume five or more liters per year, spend over 20 EUR per month, and buy high-quality extra virgin olive oil with protected origin. According to their socio-demographic features, the majority were ether retired or employees with a monthly income 1001–3000 EUR, employed as entrepreneurs, self-employed, or managers (more often than in the other two clusters). The group had slightly more participants above forty-one years of age, with occupations as rentiers, housewives, and retired persons. In all categories of olive oil attributes (health properties and sensorial, extrinsic, and intrinsic attributes of olive oil), they rated all features much higher (compared to the other two clusters). They rated the notion of olive oil as a natural product that benefits health, improves blood vessel health, improves heart health, and lowers cholesterol most highly (grades above 4). Their self-estimated knowledge about olive oil was more highly rated (3.49) then for those in the other two clusters (2.88/2.37).

The Cluster 2—moderate, utilitarian/healthy, and extrinsic consumers—consumed two liters per year, using it at home with a meal several times per month. According to their socio-demographic features, the majority were employees who had finished secondary education and whose income ranged 1001–2000 EUR monthly. While they valued the extrinsic, intrinsic, and sensorial aspects of olive oil quite highly, their perception of its health benefits was much lower than that of Cluster 1 (as discussed above). In particular, they evaluated the hedonistic element, pleasant smell of olive oil (3.94), and the feature that olive oil improves the taste and smell of food (3.86) highly. Cluster 2 perceived that olive oil is not an especially important part of their diet (2.18) and that their knowledge was lower than average (2.71). They preferred the yellow oil color (3.20) more compared to the green color (2.89), and rated the smell and the taste of green olives (3.45) compared to the smell and the taste of the ripe fruit of the olive (3.44) almost equally.

The Cluster 3—Occasional, low-interest, but health-conscious consumers—contained participants of younger age, below forty, with a relatively larger share of women and unemployed participants (14.7%) and the largest share of employees (71.4%) among all clusters. They evaluated the health benefits of olive oil (3.70 for olive oil is a natural product, it is beneficial to health, lowers the cholesterol in the blood) quite highly, as well as the hedonistic attributes of olive oil (pleasant taste (3.44) and smell (3.36) and certified chemical quality (analysis, free acidity (3.46). They appreciated the extrinsic qualities of PDO, country, and regional and micro-location less, which can be related to their general lower knowledge about olive oil (2.37) and lower self-estimated knowledge compared to the average consumer. They rarely ate a Mediterranean diet (2.01), and olive oil was not an important part of their diet (1.81).

**Table 7.** Indicators of clusters by socio-demographic features and consumption patterns.

| | German Olive Oil Consumers | | |
|---|---|---|---|
| **Indicator** | **Cluster 1: Frequent, Hedonistic, Healthy, and Sensorial** | **Cluster 2: Moderate: Utilitarian, Healthy, and Extrinsic** | **Cluster 3: Occasional, Low Interest, but Health Conscious** |
| **Consumption L/year** | ≥5 L | 2 L | ≤1 L |
| **Consumption EUR/Month** | ≥20 EUR | 11–20 EUR | ≤10 EUR |
| **Consumption Place** | Restaurants, while travelling, at home | Restaurants, while travelling, at home | At home |
| **Frequency of Consumption** | | | |
| Very rarely Several times in a year Several times a month Several times in a week Daily Multiple times daily | 1.3% 4.8% 13.9% 43.7% 30.7% 5.6% | 3.2% 13.4% 42.6% 32.9% 7.9% 0.0% | 13.6% 42.1% 35.7% 6.3% 2.3% 0.0% |
| **How Do You Consume Olive Oil?** | | | |
| Without meal/alone With meal In other ways (cosmetic, body oil) | 2.2% 93.8% 4.0% | 1.9% 87.5% 10.6% | 3.4% 87.9% 8.7% |
| **Gender** | 63.1% F 36.9% M | 60.7% F 39.3% M | 65.7% F 34.3% M |
| **Age** | | | |
| 20–30 | 11.4% | 15.2% | 14.9% |
| 31–40 | 22.5% | 24.5% | 38.6% |
| 41–50 | 31.0% | 28.7% | 29.7% |
| 51–60 | 23.0% | 22.6% | 3.0% |
| ≥61 | 12.1% | 9.0% | 13.8% |
| **Education** | | | |
| Primary (8 years) | 8.9% | 6.8% | 6.6% |
| Secondary (+4 years) | 42.5% | 53.7% | 51.2% |
| University (+5 years) | 48.6% | 39.5% | 42.2% |
| **Employment** | | | |
| Entrepreneur | 7.1% | 5.4% | 2.2% |
| Self-employed | 7.9% | 4.2% | 4.4% |
| Manager | 11.5% | 10.2% | 4.4% |
| Employee | 54.9% | 63.4% | 71.4% |
| Unemployed | 0.8% | 1.2% | 14.3% |
| Retired | 8.9% | 7.2% | 2.2% |
| Pupil/student | 3.9% | 4.2% | 1.1% |
| Other (rentiers, housewives) | 5.2% | 4.2% | 0.0% |
| **Monthly personal income** | | | |
| ≤1000 EUR | 15.7% | 23.2% | 17.6% |
| 1001–2000 EUR | 26.7% | 44.9% | 43.2% |
| 2001–3000 EUR | 31.8% | 20.3% | 23.0% |
| 3001–4000 EUR | 14.2% | 6.5% | 9.5% |
| ≥4001 EUR | 11.6% | 5.1% | 6.7% |

**Table 8.** Indicators of German olive oil consumer clusters.

| Indicator | Cluster 1: Frequent, Hedonistic, Healthy, and Sensorial | Cluster 2: Moderate, Utilitarian, Healthy, and Extrinsic | Cluster 3: Occasional, Low Interest, but Health Conscious |
|---|---|---|---|
| **Health Properties** | **M** | **M** | **M** |
| Because olive oil is a natural product | 4.17 | 4.03 | 3.70 |
| It is beneficial to health | 4.07 | 3.62 | 3.70 |
| Improves blood vessel health | 4.06 | 3.63 | 3.30 |
| Improves heart health | 4.06 | 3.66 | 3.40 |
| Lowers cholesterol in the blood | 4.03 | 3.51 | 3.70 |
| Olive oil improves the taste and smell of food | 3.99 | 3.86 | 3.20 |
| Olive oil has a positive overall effect on health | 3.77 | 3.42 | 3.50 |
| **Knowledge about Olive Oil** | | | |
| I have a lot of knowledge about olive oil | 3.49 | 2.88 | 2.37 |
| I know more about olive oil than the average consumer | 3.19 | 2.71 | 2.53 |
| Olive oil is an important part of my diet | 2.60 | 2.18 | 1.81 |
| I am eating a Mediterranean diet | 2.55 | 2.27 | 2.01 |
| **Sensorial Attributes of Olive Oil** | | | |
| The smell and taste of the olive fruit | 3.64 | 3.07 | 2.83 |
| The smell and the taste of the ripe fruit of the olive | 3.64 | 3.44 | 3.02 |
| The smell and the taste of green olives | 3.60 | 3.45 | 3.12 |
| Scent and flavor of aromatic herbs (e.g., rosemary, oregano) | 3.28 | 3.10 | 2.95 |
| The smell and the taste of green leaves and grass | 3.27 | 2.97 | 2.84 |
| **Extrinsic Attributes—PDO, Quality Labels, Certificates** | | | |
| Region (e.g., Istria, Dalmacija) | 4.17 | 3.51 | 3.30 |
| Country of origin | 3.84 | 3.62 | 3.30 |
| Protected designations of geographical origin/origin designation (PDO) | 3.75 | 3.54 | 3.24 |
| Manufacturer/brand name | 3.47 | 3.25 | 3.21 |
| Micro-localities (e.g., Poreč, Rovinj, islands, e.g., Krk, Brač) | 3.31 | 3.09 | 2.96 |
| Awards (medals) | 3.11 | 3.04 | 3.14 |
| **Hedonistic Attributes of Olive Oil** | | | |
| Pleasant smell | 4.00 | 3.94 | 3.36 |
| Pleasant taste | 4.03 | 3.76 | 3.44 |
| Certified chemical quality (analysis, free acidity) | 3.72 | 3.74 | 3.46 |
| Packaging Type (Glass) | 3.71 | 3.61 | 3.26 |
| **Intrinsic Attributes for Olive Oil** | | | |
| Color (yellow) | 3.46 | 3.20 | 2.85 |
| Bitterness | 3.45 | 3.39 | 2.79 |
| Color (green) | 3.44 | 2.89 | 3.07 |
| Piquancy | 3.42 | 3.37 | 2.83 |
| Recommendations of friends | 3.36 | 3.15 | 2.94 |
| Recommendations of vendors (producer, shop) | 3.30 | 3.17 | 2.85 |

Furthermore, the procedure of cluster analysis was used to identify clusters among the UK olive oil consumers. Three clusters were identified, which were named: (1) Frequent consumers, hedonistic and healthy; (2) occasional consumers, healthy, hedonistic, and intrinsic; (3) low interest with low knowledge. Their sociodemographic features were described through age, education, employment, income, while consumption patterns were described by place and volume of consumption and expenditures (Table 9). Indicators of clusters were noted in Table 10.

The Cluster 1 contained half of the participants in the survey (50%), while the other two contained a quarter (25%). The first cluster—frequent consumers, hedonistic and healthy—consume five or more liters per year, consume olive oil several times a week, spend over 20 EUR per month on olive oil, and buy high-quality extra virgin olive oil with protected origin. According to their socio-demographic features, the majority were either employees, managers, or self-employed, with a monthly income above 3001 EUR. In all categories of olive oil attributes (health properties and sensorial, extrinsic, and intrinsic attributes of olive oil), they rated all features much higher (compared to the other two clusters). They rated the notion of olive oil as a healthy product that improves the taste and smell of food (4.20) and benefits health (4.20), improves heart health (4.14), and improves blood vessel health

(4.02) most highly. They appreciated the hedonistic attributes, pleasant smell (4.18), and pleasant taste (3.96). Their self-estimated knowledge about olive oil was low (2.2).

**Table 9.** Indicators of UK olive oil consumer clusters by socio-demographic features and consumption patterns.

| | UK Olive Oil Consumers | | |
|---|---|---|---|
| **Indicator** | **Cluster 1: Frequent Hedonistic, and Healthy** | **Cluster 2: Occasional, Healthy, Hedonistic, and Intrinsic** | **Cluster 3: Low Interest with Low Knowledge** |
| **Consumption L/year** | ≥5 L | 1–2 L | ≤1 L |
| **Consumption EUR/month** | ≥ 20 EUR | 11–20 EUR | ≤ 10 EUR |
| **Consumption place** | At home (92.7%) | At home (94.6%) | At home (98.2%) |
| **Frequency of consumption** | | | |
| Very rarely | 3.7% | 2.0% | 7.7% |
| Several times in a year | 9.2% | 9.8% | 35.1% |
| Several times a month | 13.0% | 25.5% | 17.9% |
| Several times in a week | 53.7% | 56.9% | 20.5% |
| Daily | 13.0% | 1.9% | 17.9% |
| Multiple times daily | 7.4% | 3.9% | 0.0% |
| **How do you consume olive oil?** | | | |
| Without meal/alone | 0.0% | 3.7% | 4.0% |
| With meal | 93.6% | 92.6% | 96.0% |
| In other ways (cosmetic, body oil) | 6.4% | 3.7% | 0.0% |
| **Gender** | 44.6% F 55.4% M | 45.1% F 54.9% M | 55.6% F 44.4% M |
| **Age** | | | |
| 20–30 | 13.5% | 27.5% | 7.0% |
| 31–40 | 23.1% | 21.6% | 32.6% |
| 41–50 | 34.6% | 27.5% | 30.2% |
| 51–60 | 19.2% | 7.8% | 20.9% |
| ≥61 | 9.6% | 15.7% | 9.3% |
| **Education** | | | |
| Primary (8 years) | 0.0% | 0.0% | 2.3% |
| Secondary (+4 years) | 43.6% | 29.4% | 39.5% |
| University (+5 years) | 56.4% | 70.6% | 58.2% |
| **Employment** | | | |
| Entrepreneur | 5.5% | 2.0% | 4.7% |
| Self-employed | 14.5% | 25.5% | 11.5% |
| Manager | 23.6% | 13.7% | 25.6% |
| Employee | 47.3% | 49.0% | 44.2% |
| Unemployed | 0.0% | 0.0% | 0.0% |
| Retired | 9.1% | 2.0% | 9.3% |
| Pupil/student | 0.0% | 7.8% | 0.0% |
| Other (rentiers, housewives) | 0.0% | 0.0% | 4.7% |
| **Monthly personal income** | | | |
| ≤1000 EUR | 9.4% | 14.6% | 11.4% |
| 1001–2000 EUR | 13.2% | 16.7% | 20.0% |
| 2001–3000 EUR | 20.8% | 35.4% | 40.0% |
| 3001–4000 EUR | 34.0% | 14.6% | 14.3% |
| ≥4001 EUR | 22.6% | 18.8% | 14.3% |

**Table 10.** Indicators of UK olive oil consumer clusters.

| Indicator | Cluster 1: Frequent, Hedonistic, Sensorial, and Healthy | Cluster 2: Occasional, Hedonistic, and Healthy | Cluster 3: Low Interest, Sensorial, Low Knowledge |
|---|---|---|---|
| **Health Properties** | **M** | **M** | **M** |
| Olive oil improves the taste and smell of food | 4.20 | 3.73 | 2.87 |
| It is beneficial to health | 4.20 | 3.79 | 3.36 |
| Improves heart health | 4.14 | 3.73 | 3.47 |
| Improves blood vessel health | 4.02 | 3.65 | 3.33 |
| Because olive oil is a natural product | 4.00 | 3.93 | 3.33 |
| Lowers cholesterol in the blood | 3.88 | 3.47 | 3.33 |
| Olive oil has a positive overall effect on health | 3.82 | 3.79 | 3.36 |
| **Knowledge about Olive Oil** | | | |
| Olive oil is an important part of my diet | 2.79 | 2.10 | 1.89 |
| I am eating a Mediterranean diet | 2.52 | 2.04 | 1.51 |
| I know more about olive oil than the average consumer | 2.21 | 1.98 | 1.55 |
| I have a lot of knowledge about olive oil | 2.04 | 2.00 | 1.58 |
| **Sensorial Attributes of Olive Oil** | | | |
| The smell and the taste of green leaves and grass | 3.59 | 3.34 | 3.18 |
| Scent and flavor of aromatic herbs (e.g., rosemary, oregano) | 3.55 | 3.44 | 2.98 |
| The smell and taste of the olive fruit | 3.45 | 3.30 | 3.27 |
| The smell and the taste of the ripe fruit of the olive | 3.13 | 3.26 | 3.20 |
| The smell and the taste of green olives | 3.11 | 3.12 | 3.22 |
| **Extrinsic Attributes—PDO, Quality Labels, Certificates** | | | |
| Country of origin | 3.63 | 2.98 | 2.72 |
| Manufacturer/brand name | 3.36 | 2.82 | 2.89 |
| Protected designations of geographical origin/original design | 3.13 | 3.27 | 2.88 |
| Region (e.g., Istria, Dalmacija) | 2.93 | 2.82 | 2.62 |
| Micro-localities (e.g., Poreč, Rovinj, islands, e.g., Krk, Brač) | 2.92 | 2.76 | 2.53 |
| Awards (medals) | 2.73 | 3.30 | 3.20 |
| **Hedonistic Attributes of Olive Oil** | | | |
| Pleasant smell | 4.18 | 3.98 | 3.58 |
| Pleasant taste | 3.96 | 3.96 | 3.80 |
| Packaging type (Glass) | 3.55 | 3.38 | 3.09 |
| Certified chemical quality (analysis, free acidity) | 3.52 | 3.44 | 3.36 |
| **Intrinsic Attributes for Olive Oil** | | | |
| Recommendations of friends | 3.41 | 3.46 | 3.18 |
| Color (yellow) | 3.14 | 3.04 | 3.02 |
| Color (green) | 3.04 | 3.10 | 2.80 |
| Bitterness | 3.00 | 3.32 | 2.89 |
| Piquancy | 2.93 | 3.14 | 2.91 |
| Recommendations of vendors (producer, shop assistant) | 2.88 | 3.14 | 2.91 |

Compared to the German sample, the first and second clusters of the UK sample contained more men, more managers and self-employed, lower age (below 40 years), and better-educated participants with higher income. They appreciated the health properties and pleasant smell of olive oil much more compared to Germans.

The second cluster—occasional, healthy, hedonistic, and intrinsic—consumed, on average, a very moderate amount of olive oil (31.4% used two liters, and 68.6% used one liter per year) at home several times a month or week. According to their socio-demographic features, the majority were either employees or self-employed, with an average monthly income of 2001–3001 EUR. Compared to the other two clusters, cluster 2 had younger participants, more with a university education, and fewer managers and retired persons. They appreciated bitterness (3.32), awards (3.30), piquancy (3.14), PDO (3.27), and recommendations of friends (3.46) and vendors (3.14) much more compared to the other two clusters. They preferred green color (3.10) more than yellow color (3.04), but preferred the ripe smell and taste (3.26) more than that of the green olives (3.12). Their self-estimated knowledge (2.0) and average knowledge (1.98) about olive oil was low.

The third cluster—low interest with low knowledge—consumed the lowest amount of olive oil (below one liter per year), at home, several times a year. Compared to the other two clusters, Cluster 3 had the youngest participants, and was similar to Cluster 1 in education and the share of entrepreneurs, managers, and retired persons. Their average monthly income was 2001–3001 EUR.

They appreciated the health properties of olive oil with its benefits to health (3.36) and heart health (3.47). In addition, they appreciated the hedonic attributes, such as pleasant taste (3.80) and pleasant smell (3.58), as well as extrinsic attributes like awards (3.20) and intrinsic attributes like the smell and taste of olive fruit (3.27), yellow color (3.02), piquancy (2.91), and recommendations of friends (3.18) and vendors (2.91). Their self-estimated knowledge (1.55) and average knowledge (1.58) about olive oil was very low.

## 4. Discussion and Conclusions

Our survey estimated that both German and UK olive oil consumers will share some similarities in their behavior towards olive oil. The UK sample contained more men and younger participants with higher education and monthly income. The socio-demographic features of participants in the survey showed a significant impact on behavior toward olive oil consumption (frequency and volume).

Both of these countries have traditionally had no olive oil production or cultural heritage and tradition of consumption; therefore, we expected that their behavior toward olive oil might be different from or even opposite to the traditions in Mediterranean countries. Due to this fact, the survey participants expressed no family traditions in consumption and that olive oil was not a part of their culture.

Their main preferences were related to the health properties of olive oil, followed by the hedonistic attributes. Health properties were very important among both samples and all clusters of participants, and were especially highly valued by the highly educated and high-income segment (similarly to [21]).

Through the differentiation with cluster analysis, we came to three different types of German consumers, which were named (1) Frequent—hedonistic, healthy, and sensorial; (2) Moderate—utilitarian, healthy, and extrinsic; and (3) Occasional—low-interest but health-conscious olive oil consumers.

The three clusters were different considering their average appreciation of health attributes; the German sample valued all attributes in the Frequent cluster more highly with the exception of the statements that olive oil improves the taste and smell of food and that it is beneficial to health and heart health. Our findings can be matched with the surveys of [21,25,31] with clusters based on the health attributes of olive oil. The health properties of olive oils were very highly valued among all three clusters; therefore, in marketing strategies it should be promoted as a healthy product that is beneficial to heart and blood vessel health and that lowers cholesterol. In our findings, we can confirm that persons above forty years of age and with higher income were more aware of olive oil's health properties (similarly to [19]); therefore, the health aspect should be especially marketed to the healthy and hedonic cluster of German consumers.

The attributes of olive oil—sensorial, extrinsic, and intrinsic attributes—were almost equally important, with scores highly saturating the factor matrix.

The hedonistic attributes of olive oil (pleasant smell and pleasant taste) and the sensorial attributes (the smell and the taste of green olives and the smell and the taste of the ripe fruit of the olives) should be highlighted in German marketing strategies.

The perspective of the German market has great potential for Croatian olive oils, since the surveyed consumers evaluated the regional and micro-localities of production highly in the context of Croatia as a country of production, similarly to the findings of [10,13], whereas country and region of production can be used in marketing strategies. For local Croatian farmers, the importance of local production for German consumers assures the sustainability of local production.

Official cues (labels of protected geographic origin (PDO labels)), origin cues (country of origin, region of origin), and sensory cues (color, appearance, price, packaging) were perceived as assurances of the high quality level and originality of the product (similarly to the findings of [8]), and can assure

sustainability connected to local production, which has distinctive features of olive oil (similarly to the findings of [22,30]).

The differentiation of the low-interest cluster confirms the findings [2]; due to their lack of knowledge about olive oil as a product, we cannot expect much differentiation in this segment of consumers. We might propose to highlight the country and region of production and health benefits and to offer oils with less piquancy and bitterness to avoid their (dis)preference [31]. Therefore, we can use the strategy mentioned by [28]. Further differentiation might be expected for the hedonic and utilitarian cluster, for whom extrinsic, intrinsic, and sensorial attributes should be used.

Similarly to the German participants in the survey, the UK participants appreciated the health properties and hedonic attributes of olive oil. Their validation of attributes was somewhat lower and as was their knowledge.

Their favorite way of consumption was at home with a meal in all segments. In addition, we noticed that the intrinsic attributes of bitterness and piquancy were the least appealing attributes, similar to the findings of dis-preference for bitterness by [31].

Since consumption of olive oil was not a part of their daily diet and was not a habit, and because UK consumers did not follow a Mediterranean diet, the efforts of marketing olive oil should be much stronger than in the Mediterranean countries with traditional and culturally established habits of consumption. Therefore, new concepts of knowledge transfer about olive oils should be introduced, e.g., introducing the value of "terroir" (similarly to wine [32]), transferring scientific knowledge to the open public, and increasing the "culture of olive oil and promoting producers and oleo tourism through the tourism system" [33].

In both samples, all clusters mentioned glass as the preferred type of packaging. It was considered a hedonic attribute connected to the image of high quality (which differed from preference for plastic found in [9]).

Through cluster analysis, we came to three clusters with differences in behavior: Cluster 1—frequent, hedonistic, sensorial, and healthy; Cluster 2—occasional, hedonistic, and healthy; Cluster 3—low-interest, sensorial, and low-knowledge consumers.

The frequent cluster appreciated all attributes more compared to the other two clusters, with the exception of green and ripe olive oil smell and taste, while bitterness and piquancy were most appealing to the cluster of occasional users.

The factor analysis showed high saturation of all variables in only two factors, and pointed out that UK consumers rely highly on the vendors' recommendations because their self-estimated knowledge was pretty low. Therefore, the role of local olive oil producers and other vendors is important in the long run, and their knowledge is valuable in terms of local sales.

The extrinsic attributes, such as region, micro-locality, and PDO, were highly valued and could lead the UK consumer toward a decision in the purchase process (along with the vendors' guidance), also enhancing economic sustainability for local producers.

Since the sensorial attributes were highly valued, such as smell and taste of green grass and leaves, aromatic herbs, and olive fruits that are very particular for certain micro-localities, marketing strategies can highlight this aspect in the promotion of certain oils to certain segments of consumers in the UK. Although they preferred a green smell and scent, they appreciated a yellow more than green olive oil color (similar to findings in Spain, [18]).

Given the fact that benefits to health in general and health of the heart and blood vessels were highly appreciated by UK consumers, these features could be used by marketers in their sales and promotion activities.

Although our survey was representative, we reached some limitations due to financial constraints; therefore, further research is expected as a follow up of this survey through a wider cross-national survey of consumer behavior toward olive oil and including a wider range of stakeholders (producers, vendors, small farmers, and small and medium enterprise producers) to measure multiplicative effects in production and sales that might enhance economic sustainability.

**Funding:** This research received no external funding; the author acknowledges that this research was supported by personal financial means.

**Acknowledgments:** We would like to express our gratitude to the olive oil consumers who patiently answered our questions during their stay in Istria in the period from June to September 2019. We owe special gratitude to hotel companies of Laguna and Valamar in Poreč and Umag, Maistra in Rovinj, and Aminess in Novigrad, their management that approved our survey, and the hotel management and staff who helped on site. The author expresses gratitude to Ana Težak Damijanić for organizing contacts with hotel companies and management and to Marija Pičuljan mag.oec. for assistance in handing out questionnaires. The author would like to thank Chinicci Gaetano, Mesquita Daniel, Sirieux Lucie, and Hely Tomila for sharing their newest survey results.

**Conflicts of Interest:** The author declares no conflict of interest.

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
