# Peer review of "Segmenting Olive Oil Consumers Based on Consumption and Preferences toward Extrinsic, Intrinsic and Sensorial Attributes of Olive Oil"

_sustainability, doi:10.3390/su12166379_

Round 1

Reviewer 1 Report

This article presents a topic of great importance for olive oil consumers and producers. However, throughout the document, some weaknesses were identified that make the work somewhat poor.

Introduction

The introduction is well structured and has a logical chronology. In this session, there is a great importance to characterize the problem that the work wants to help answer. In this context it is well defined. However, in the summary it is described that the study work is about German and United Kingdom consumers and in the introduction, the objective is only focused on German consumers. All information must be in agreement.

Material and methods

In the material and methodology applied to the development of the work there is information that has no scientific importance, namely the way the material was distributed (pens, test papers, etc.) and the way they were identified. There is a lack of more detailed information such as what types of oils were used for the tasting, how the sensory tasting was performed (with the help of some food or if it was consumed directly without any product), the type of commercial category, the intensity fruity (ripe, soft green and intense green). All of this information is important and should be described to better understand the trend of preferred consumers. Regarding the used proof sheet, it is necessary to mention which model was used and which parameters were evaluated. Another essential point is what kind of test was performed? sensory proof of the descriptive profile? hedonic proof? since they are consumers and not semi or training tasters, it is necessary to use a sheet that is easy for consumers to interpret. This information is not described in the material and methods. The scale used (1-11) is not the scale normally used by the IOC. They only use this scale in table olives. For oils the scale used is 0-10.

Results discussion and conclusion

Regarding the results obtained, it clearly shows that there are differences in the evaluated attributes. Some of them with high weight. However, the authors could use other types of statistical tools, namely PCA, LDA in order to obtain a more robust treatment and information. In addition, these statistical tools would allow to better separate the intrinsic and extrinsic factors.

Author Response

  1. Open Review

Thank You for taking Your time and for Your effort in reviewing the paper.

Introduction

The introduction is well structured and has a logical chronology. In this session, there is a great importance to characterize the problem that the work wants to help answer. In this context it is well defined. However, in the summary it is described that the study work is about German and United Kingdom consumers and in the introduction, the objective is only focused on German consumers. All information must be in agreement.

change:

Authors mistake in writing, the UK consumers are now mentioned too, now everywhere.

For UK, example, literature 9, 11

The summary was changed with improved text changes in reserach results and mention of finding in segments of both samples, possible marketing strategies and which attributes can be highlighted.

Material and methods

In the material and methodology applied to the development of the work there is information that has no scientific importance, namely the way the material was distributed (pens, test papers, etc.) and the way they were identified. There is a lack of more detailed information such as what types of oils were used for the tasting, how the sensory tasting was performed (with the help of some food or if it was consumed directly without any product), the type of commercial category, the intensity fruity (ripe, soft green and intense green). All of this information is important and should be described to better understand the trend of preferred consumers. Regarding the used proof sheet, it is necessary to mention which model was used and which parameters were evaluated. Another essential point is what kind of test was performed? sensory proof of the descriptive profile? hedonic proof? since they are consumers and not semi or training tasters, it is necessary to use a sheet that is easy for consumers to interpret. This information is not described in the material and methods. The scale used (1-11) is not the scale normally used by the IOC. They only use this scale in table olives. For oils the scale used is 0-10.

change:

The methodology was described in detail and therefore the mention of technical details. Threfore the sentence about distribution of means.

The author agree thatit has little sceintific value and therefore it was erased.

The author supposes a misunderstanding of the test taken, because there was no actual olive oil tasting on site due to lack of finance, time and organization – technical side of olive oil tasting. It would be technically imposible for two persons to do it in hotels with a thousand people (each participant should have an isolated cabine with heating equipment, special tasting glases, and be guided while tasting etc.).

The ratings for all attributes were stated by participants as opinions about each attribute rated from 1 to 5.

The IOC scale is rated 1-10 (and mentioned in literature reference 26) while in the questionare we used a more suitable 1-5 based on Likert scale.

In the future there are plans to upgrade the research and apply for a new project which will contain also consumers behaviour and which will have enough financial means to finance oil tastings with a larger number of consumers.

Results discussion and conclusion

Regarding the results obtained, it clearly shows that there are differences in the evaluated attributes. Some of them with high weight. However, the authors could use other types of statistical tools, namely PCA, LDA in order to obtain a more robust treatment and information. In addition, these statistical tools would allow to better separate the intrinsic and extrinsic factors.

change:

We are greatfull for the suggestion and might use it in future data processing. We wanted to collect as much possible data about consumers in non producing countries and therefore we might have put all attributes toghether to get an overwiew of the all. Also wewanted to use the most appropriate method that would allow at the same time  acomparison of all data in both samples. Therefore e used factor analisys and segmentation through cluster analysis.

We expect to create some regression models and use other techniques as well in a future paper, especcialy for UK consumers for whom factor analysis showed a dense result.  

Since we were creating this research for the first time and we did not have much insight before the actual survey we are planning to make some changes in the future surveys, hopefully in 2021, since this year 2020we have very few foreign vistors in hotels and therefore a survey cannot be executed.  

Reviewer 2 Report

The paper is very interesting for me.
I think that is necessary only uniform the decimal numbers in all manuscript.

Author Response

  1. Open Review

Thank You for taking Your time and effort in reviewing the paper.

I think that is necessary only uniform the decimal numbers in all manuscript.

change:

According to the suggestion all numbers are checked in the manuscript.

Beside the all the numbers in tables are now set in centre.

Also according to other two reviewers some minor changes are made in the summary – added research results, in introduction – added literature from Sustainability review and literature for UK olive oil consumers, a notion of sustainability is added in discussion too.  

Reviewer 3 Report

The manuscript concerns a segmentation analysis of olive oil consumers from the UK and Germany.  The analysis seems well performed, the sample sizes are large, and the introduction is very helpful.  Despite this, there remain a few issues that warrant attention:

  • The abstract is not particulalrly informative, it is hard to discern what data the researchers performed segmentation on, or gain any information on the segments themselves.
  • The references in many sections (i.e. line 34) are referred to in a strange and non-grammatical manner which I’ve never seen before.
  • Acronyms need to be defined
  • If, as stated in the intro, “The purpose of this paper was to find out how German olive oil consumers differ according to 76 their consumption habits and preferences then why were UK consumers also used?
  • Any study involving human subjects needs both a statement on the study’s approval by an institutional review board, and also details on the consent procedure.
  • Line 162 “hart”
  • Grammar throughout needs some attention
  • There are as I can understand no references at all to the topic of sustainability in the manuscript at all, in fact the word doesn’t not appear at all outside of the journal header.

Author Response

  1. Open Review

Dear reviewer,

Thank You for taking Your time and effort in reviewing the paper.

Comments and Suggestions for Authors

The manuscript concerns a segmentation analysis of olive oil consumers from the UK and Germany.  The analysis seems well performed, the sample sizes are large, and the introduction is very helpful.  Despite this, there remain a few issues that warrant attention:

  • The abstract is not particulalrly informative, it is hard to discern what data the researchers performed segmentation on, or gain any information on the segments themselves.

changes:

The abstract was changed with additional explanations of research results and the segments of both surveyed samples. The aspect of sustainabilty for both samples are pointed out too.  

  • The references in many sections (i.e. line 34) are referred to in a strange and non-grammatical manner which I’ve never seen before.

change:

The references are double checked and corrected, the author has noticed several flaws.

  • Acronyms need to be defined
  • If, as stated in the intro, “The purpose of this paper was to find out how German olive oil consumers differ according to 76 their consumption habits and preferences then why were UK consumers also used?

changes:

Authors mistake in writing, the UK consumers are now mentioned too, now everywhere.

Literature was adjusted , for emaple for UK consumers nr. 9.

  • Any study involving human subjects needs both a statement on the study’s approval by an institutional review board, and also details on the consent procedure.

mentions:

The survey was connected to the authors decade of work in the Tasting Panel for olive oil at the Institute of agriculture and toursim in Poreč, Croatia.

The idea to examine consumers behviour came from the work on Horizon project Oleum 2017-2020 where the author takes part as researcher. Since this project had no work package about consumers of olive oil the author financed the idea and the survey by own finnancial means. And therefore we cannot mention the Horizon project.

The research was approved by the institution director where the author was employed, the  questionnaire was also approved by the institution and Panel Head. Finally it was aproved by hotel company managers.

  • Grammar throughout needs some attention
  • Line 162 “hart”

changes:

Gramatical errors were corrected in the text.

  • There are as I can understand no references at all to the topic of sustainability in the manuscript at al, in fact the word doesn’t not appear at all outside of the journal header.

changes:

The notion of sustainabilty was added in abstract, introduction and discussion and conclusions. The latest literature from Sustainability review was used (2019/2020). Four papers were used.

The author suspected if the use of the Sustainability is allowed in the text, since some journals hesisate to self cite their texts while other journals encourage it.  

Therefore other journals were used from the same company MDPI; Foods

Sustainability aspects are now mentioned in literature also by 22,23,24,25 reference

Reviewer 4 Report

The topic is not really the most interesting for the journal, I think that most appropriate will be a journal of food product marketing.

The segmentation of consumers, olive oil consumption as a product, and the preferences are kind of consumer behavior and I cannot see the connection to the journal.

I couldn't help my self noticing the very well structured methodology research, the cluster analysis, and the good presentation of results.

Interesting results for this kind of "international" research on german and UK respondents.

Author Response

Dear reviewer,

thank You for Your time and effort toenhance the article.

we have made following changes

we mentioned adequate conection with aded text in the light of Sustainability

- aspects for producers, especcially small farmers

  • aspects for local production
  • aspects for markets

we have set 4 references from 2019 and 2020 from the journal Sustainability to enhance our statements in the text.

we have made some changes in the language since minor errors were detected

conclusions were changed with additional explanations and added aspect of sustainability

Round 2

Reviewer 1 Report

After the authors' corrections, I consider that the article has the requirements to be published.

Author Response

Thank You for Your time and effort to help me achieve a better quality of the text.

during the few days od reviews awainting some minor changes were done to achieve fluidity and equalness:

  1. in table 1. everywhere where perecntages were a whole number e.g. 46% 54% it was changed to 46.0% 54.0% to equalize all data
  2. brackets (  ) were put in table 1. at literature 12 and 17
  3. in line 52 in the sentence while (16) found the change is: while for Italian consumers (16) found...
  4. line 56 cultural differences in food habits is mentioned consumption habits therefore changed
  5. line 57 was put in brackets ( even different cultures in the same country...)
  6. in line 81 sustainability in production... olive oil was put since it consider olive oil
  7. in line 111 words .. we gather .. were erased
  8. in lines 209-211 and 226-227 232 small leters insteads of capital were put for attributes protected, original and olive, colour, bitternes, piquancy, recommendations
  9. due to automatic computer work after the point a capital letter was erased in line 293, the cluster 1 Contained
  10. in line 308 hedonistoic and healthy were set vice versa
  11. the limitations of the study and future surveys were set at the end of the paper